# Effective PSCCH Searching for 5G-NR V2X Sidelink Communications

Roberto Magueta [1,*] , João Domingues [2], Adão Silva [2] and Paulo Marques [1,3]

1   Allbesmart LDA, Avenida do Empresário, Centro de Empresas Inovadoras 1,
    6000-767 Castelo Branco, Portugal; pmarques@allbesmart.pt
2   Instituto de Telecomunicações (IT) and Departamento de Eletrónica, Telecomunicações e Informática (DETI),
    University of Aveiro, 3810-193 Aveiro, Portugal; joaofdomingues@ua.pt (J.D.); asilva@av.it.pt (A.S.)
3   Instituto Politécnico de Castelo Branco (IPCB), Avenida Pedro Álvares Cabral 12,
    6000-084 Castelo Branco, Portugal
*   Correspondence: rlm@ua.pt

**Abstract:** Cooperative Intelligent Transport Systems (C-ITS) are essential for increasing road safety and to make road transport more efficient, sustainable, and environmentally friendly. The implementation of C-ITS technology is only possible through the connectivity of Vehicle-to-Everything (V2X), which allows the interconnection of vehicles in a network and with road support infrastructure. However, real-time systems require efficient signal processing in order to respond within the necessary time. Some of this processing is related to searching the Physical Sidelink Control Channel (PSCCH), where a blind algorithm is commonly used. However, this algorithm is quite inefficient to searching the PSCCH, since all the processing should be completed several times before successful decoding it. Therefore, the aim of this paper is to design a more efficient algorithm to search/decode the PSCCH. In the proposed algorithm, we firstly compute all the correlations between the received signal and the Demodulation Reference Signal (DMRS), and the remaining conventional processing to decode the PSCCH is only performed over the subchannels with higher correlation, which leads to a strong complexity reduction. The proposed algorithm is evaluated and compared with the conventional blind algorithm. The results have shown a significant performance improvement in terms of runtime.

**Keywords:** V2X; sidelink; 3GPP; 5G-NR; PSCCH; real-time systems





## 1. Introduction

Vehicle networks are seen as a key enabler for connected and autonomous driving, but have been limited to increase the safety of road users and improve the traffic efficiency of roads [1]. However, the Fifth-Generation (5G) New Radio (NR) V2X comes to support new applications, where ultra-low latency and very high reliability are required, and can meet an end-to-end latency of 3 ms with a high reliability up to 99.999% [2]. The 5G-NR V2X will not replace its Long-Term Evolution (LTE) version, i.e., the cellular V2X (C-V2X), but they might coexist, giving support to the use cases that cannot be supported by C-V2X [3]. The requirements for both periodic and aperiodic traffics, as well as the half-duplex and the hidden node problem, are the main aspects for 5G-NR V2X physical layer design [4]. One of the main new features of 5G-NR V2X physical layer is to support the different subcarrier spacing associated to different frequency range, which enables robustness against Doppler [5]. However, there are studies on the impact of numerology on V2X Communications which claim that increasing subcarrier spacing involves a trade-off between the Inter Carrier Interference (ICI) and the Inter Symbol Interference (ISI) effects over the system performance [6].

In 5G-NR V2X, user equipment's (UEs) are vehicles, Road Side Units (RSUs), or mobile devices that are carried by pedestrians [7], which brings us to four different types of V2X communications: Vehicle-to-Vehicle (V2V), Vehicle-to-Infrastructure (V2I), Vehicle-to-Pedestrian (V2P), and Vehicle-to-Network (V2N) [8]. One or a combination of V2V,

V2I, V2P, and V2N may be applied to each case, which are divided in the following four groups [9]:

- Vehicles Platooning: It supports the formation of an interconnected group of vehicles which exchange information to get shorter distances and fuel saving for the platoon;
- Advanced Driving: It enables the semi-automated or fully-automated driving, where each vehicle and/or RSU shares data obtained from its local sensors with surrounding vehicles in proximity, to coordinate their trajectories or maneuvers, which increases the safety, avoids collisions, and improves the traffic efficiency;
- Extended Sensors: It allows to improve the perception of the environment beyond the perception capabilities of the drivers, through the collection of information by local sensors present in vehicles, RSUs, mobile devices of pedestrians, or V2X application servers;
- Remote Driving: It allows to control vehicle driving remotely, for dangerous and severe conditions, or for passengers who cannot drive themselves, whether by humans or cloud and edge computing applications.

The 3rd Generation Partnership Project (3GPP) standard defines two frequency bands as the operating bands for V2X: 5.9 GHz band (n47) and 2.5 GHz band (n38), where the supported channel bandwidths in both frequencies are 10, 20, 30, and 40 MHz [10]. Both frequency bands belong to Frequency Range 1 (FR1) ranging from 410 MHz to 7125 MHz, and therefore none belongs to the millimeter Wave (mmWave) Frequency Range 2 (FR2) yet, ranging from 24.25 GHz to 52.6 GHz [10]. A new performance metric is needed, because other technologies may also work in the 5.9 GHz band in some regions, which may affect the 5G-NR V2X system performance, namely the safety-critical applications [11]. Resource allocation in these bands for 5G-NR V2X is defined in two modes: Mode 1 and Mode 2 [12]. In Mode 1, the vehicles are covered by one or more base stations, and they are the ones which proceed to configured or dynamic scheduling of resources. In configured scheduling, resource allocation is based on pre-defined bitmap, while dynamic scheduling is performed every millisecond based on the varying channel conditions. In Mode 2, the UE is out-of-coverage and autonomously selects the resources from a preconfigured resource pool. There are four approaches to select the resources in Mode 2 [13]:

- Each UE selects its resources autonomously through a sensing-based semi-persistent transmission mechanism;
- A cooperative distributed scheduling is performed, where each UE assists the others to select the resources;
- The UE selects the resources based on the preconfigured scheduling;
- The UE schedules the resources for its neighbors.

The sensing-based semi-persistent transmission mechanism is quite explored, however, long intervals with the same allocation may result in persistent collisions, which reduces safety due to consecutive packet losses [14].

### 1.1. Previous Works on Resource Allocation and Detection

In [15], an approach based on the measurement of the reference signals' received power (RSRP) was described. During the sensing window, the transmitting UE measures the RSRP of all subchannels, and if a subchannel does not exceed a threshold, this subchannel becomes a candidate. Moreover, a transmitting UE needs to detect the PSCCH transmitted by other UEs and receive the Sidelink Control Information (SCI) to know which subchannels are occupied. For this purpose, it is not needed all the ELDs of SCI, so they are transmitted in two stages. The first stage of SCI is transmitted over PSCCH, while the second one is transmitted over Physical Sidelink Shared Channel (PSSCH). The first stage SCI decoding is based on a blind detection. When the first stage SCI decoding fails, the second stage SCI decoding also fails. After the sensing window, we have the selection window, where the candidate subchannels are selected and occupied. In [16], an open-source, fullstack, end-to-end, standard-compliant network simulator was presented, where the physical

layer receives the first stage SCI transmitted over PSCCH to measure the RSRP and to get the information about the possible PSSCH. The RSRP is computed using the three resource elements (REs) per resource block (RB), carrying the first stage SCI, because the simulator does not explicitly have PSCCH DMRS. In [17], the authors proposed an enhancement to the sensing-based semi-persistent scheduling to reflect the stochastic nature of the aperiodic traffic on the resource allocation, to reduce the packet collisions. The authors of [18] proposed a method for pedestrian UEs that operates with partial sensing, which is used to reduce the power consumption, in order to enhancing the throughput. For this, an additional UE, i.e., an RSU, is used to collect the sensing results from each UE. Then, the RSU aligns all of the partial sensing windows to provide a more efficient sensing result.

In [19], the collision probability via the random selection of resources was analyzed, and the deep reinforcement learning algorithm was investigated, which decreases the collision probability by using a platoon leader to learn from the communication environment. In [20], a group scheduling mechanism is proposed, where a platoon leader takes charge of the whole platoon. It is the platoon leader that grants the resources by the network and distributes the scheduling information among the remaining platoon.

The authors of [21], to address the latency problem, proposed the hyper-fraction channel allocation method, where the road is divided into several zones and for each zone, a channel is allocated. Then, when a UE is located within a certain zone, this one uses the channel allocated to the zone. In [22], a cluster-based resource selection scheme is proposed to reduce the resource collision. In this scheme, the resources are divided into different resource sets, then each cluster head selects its resource set based on measurement. The authors of [23] proposed a clustering-based resource management scheme for latency and sum rate optimization. First, an optimization of the total average sum rate of each link is performed, and then cluster-based optimum algorithms are proposed to obtain the optimum resource management.

In [24], a method to minimize the interference between UEs was proposed. For that, an optimization problem is expressed as a mixed binary integer nonlinear programming, subject to the quality of service, the total available power, the interference threshold, and the minimal transmission rate. A resource allocation mechanism adaptive to the environment was proposed in [25]. This approach is based on reinforcement learning, and intends to prioritize the channel access to UEs with urgent needs. The reinforcement learning framework is modeled as a contextual multi-armed bandit to provide efficiency and accuracy. The authors of [26] proposed a vehicle density-based two-stage resource management scheme to reduce latency, improve throughput, and reliability for V2V and V2N. In the first stage, the density information for the resource distribution strategy is explored, then, in the second stage, the channel state information and queuing state information are used.

The algorithms presented above are summarized in the Table 1.

**Table 1.** Algorithms proposed in the literature.

| Algorithm Group | Algorithm Description | Reference |
|---|---|---|
| Sensing based | Subchannel selection based on RSRP measurement, computed using the DMRS. | [15] |
| | Subchannel selection based on RSRP measurement, computed using the 3 REs per RB, carrying the first stage SCI. | [16] |
| | Semi-persistent subchannel selection to reflect the stochastic nature of the aperiodic traffic. | [17] |
| | Partial sensing based on an additional UE to collect the sensing results. | [18] |

**Table 1.** *Cont.*

| Algorithm Group | Algorithm Description | Reference |
|---|---|---|
| Platooning based | Subchannel selection by a platoon leader which learns from the communication environment. | [19] |
| | Subchannel selection information distributed by a platoon leader which grants the resources by the network. | [20] |
| Geographic and clustering based | When a UE is located within a certain zone, this one uses the subchannel allocated to the zone. | [21] |
| | The resources are divided into different resource sets, and each cluster head selects its resource set. | [22] |
| | Clustering-based resource management based on an optimization of the total average sum rate of each link. | [23] |
| Others | Subchannel selection to minimize the interference between UEs. | [24] |
| | Subchannel selection based on reinforcement learning to prioritize UEs with urgent needs. | [25] |
| | Subchannel selection based on the density information and channel and queuing state information. | [26] |

### 1.2. Motivations

Although there are several algorithms to reserve resources for transmission as presented above, in the receiving UE, the PSCCH searching is completed in a pure blind way, trying all the possibilities, or at most is considered some kind of threshold to try to eliminate candidates. The threshold approach is not efficient because if the threshold is low, we still have a high number of candidates, or if the threshold is high, we could be eliminating candidates that contained a PSCCH and thus deteriorating the overall system performance. In a semi-persistent subchannel selection we can perform an optimization in the PSCCH searching because allocation is constant over a period of time, however this approach can lead to persistent collisions during this period, mainly in the Mode 2 of the 5G-NR V2X [14]. Therefore, the Blind algorithm is the one that guarantees the best system performance in terms of PSCCH detection, but at the expense of longer runtime as all possibilities are tested. Then, a new algorithm with the same performance as the Blind algorithm, but with more efficient processing to reduce the runtime, is crucial for practical 5G-NR V2X systems.

### 1.3. Contributions

In this paper, we propose an effective PSCCH searching algorithm for 5G-NR V2X sidelink communications in the Mode 2. The proposed algorithm can be applied:

-   At the transmitting UE, when it performs a resource allocation based on a sensing-based algorithm;
-   At the receiving UE, when it is trying to decode the PSCCH to obtain the desired data contained in the transport block of PSSCH.

To the best of our knowledge, PSCCH searching algorithms, beyond the Blind algorithm as in [15], specifically designed for 5G-NR V2X sidelink communications have not been addressed in the literature, yet. The main contributions of this paper are the following:

-   A correlation analysis is performed to verify the peak values of the correlation of the received signal with the PSCCH DMRS, and the predominance of the desired peak over the others;
-   Based on the previous correlation analysis, an efficient algorithm to search/decode the first stage SCI in 5G-NR V2X sidelink communication systems is designed;

—   Finally, a theoretical runtime analysis is performed to compare the runtime of the proposed algorithm with the Blind algorithm, because it is the one that guarantees the best overall system performance in terms of PSCCH detection.

The simulation results show that the Proposed algorithm achieves the same performance as Blind algorithm in terms of PSCCH detection, SCI Block Error Ratio (BLER) and throughput, but with a much shorter runtime.

The remainder of this paper is organized as follows: Section 2 describes the system model considered in this work. Section 3 presents the design of the proposed algorithm. Finally, the main performance results are shown in Section 4 and the conclusions in Section 5.

### 1.4. Notations

A matrix $\mathbf{A}$ is denoted by boldface capital letters and the column vector $\mathbf{a}$ is denoted by boldface lowercase letters. The operator $(.)^T$ denotes the transpose of a matrix. The functions $\text{corr}(\mathbf{a}, \mathbf{b})$, $\text{sort}_{desc}(\mathbf{a})$, and $\text{rem}(c, N)$ represent the correlation between $\mathbf{a}$ and $\mathbf{b}$, the sort the vector $\mathbf{a}$ in descending order, and the remainder after division of $c$ by $N$, $c, N \in \mathbb{Z}$, respectively. The vector $\mathbf{a} = [\mathbf{a}_q]_{1 \leq q \leq Q_1} \in \mathbb{C}^{Q_1 Q_2}$ is the concatenation of vector $\mathbf{a}_q \in \mathbb{C}^{Q_2}$. Finally, the indices, $t$, $k$, $l$, $n$, and $q$ represent the time domain, subcarrier in the frequency domain, the OFDM symbol, the subchannel, and sample, respectively.

## 2. System Model

In this manuscript, we consider a 5G-NR V2X sidelink system, which uses the Cyclic Prefix-Orthogonal Frequency Division Multiplexing (CP-OFDM)—as the access technique. For the sake of simplicity and, without loss of generality, we consider only two UEs, where the transmitter and receiver UE are equipped with $N_{tx}$ and $N_{rx}$ antennas, respectively. The UEs are communicating based on a resource allocation scheme in the Mode 2, Figure 1, i.e., both UEs are out of coverage and have autonomous resource selection.

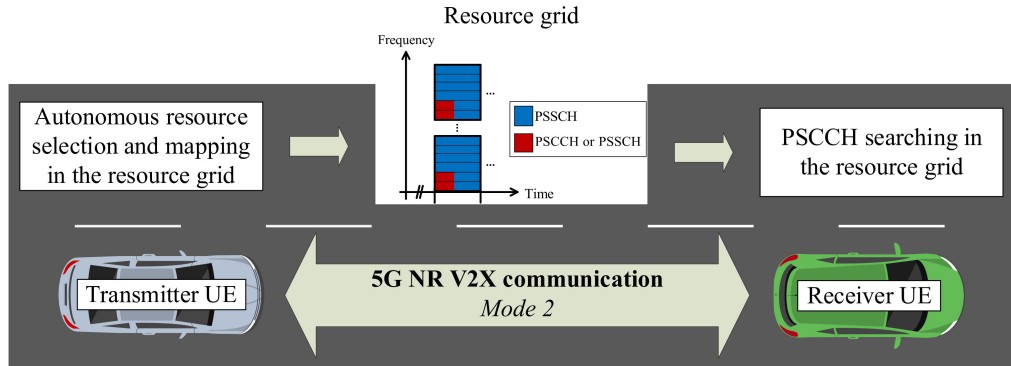

**Figure 1.** System model.

These resources are mapped in time and frequency domains as we can see in Figure 2. The smallest unit in a sidelink (SL) communication is the RE, composed by a subcarrier $k$, and an OFDM symbol $l$, where $N_{RB,SC}^{SL} = 12$ subcarriers (SC) are a RB, with a subcarrier spacing equal to $15 \times 2^\mu$ kHz, and 14 OFDM symbols corresponds to a time slot, $t_i^{SL}$, as presented in Figure 2, with $0 \leq t_i^{SL} < 10,240 \times 2^\mu$, $0 \leq i < T_{\max}$, where $\mu \in \{0, 1, 2, 3, 4\}$ defines the 5G-NR numerology [27]. In the time domain, a set of $2^\mu$ slots corresponds to 1 ms. In the frequency domain, we have $N_{subCh}^{SL}$ subchannels with $N_{subChSize}^{SL}$ RBs each one, in a total of $N_{RB}^{SL} = N_{subCh}^{SL} N_{subChSize}^{SL}$ RBs available for sidelink transmissions.

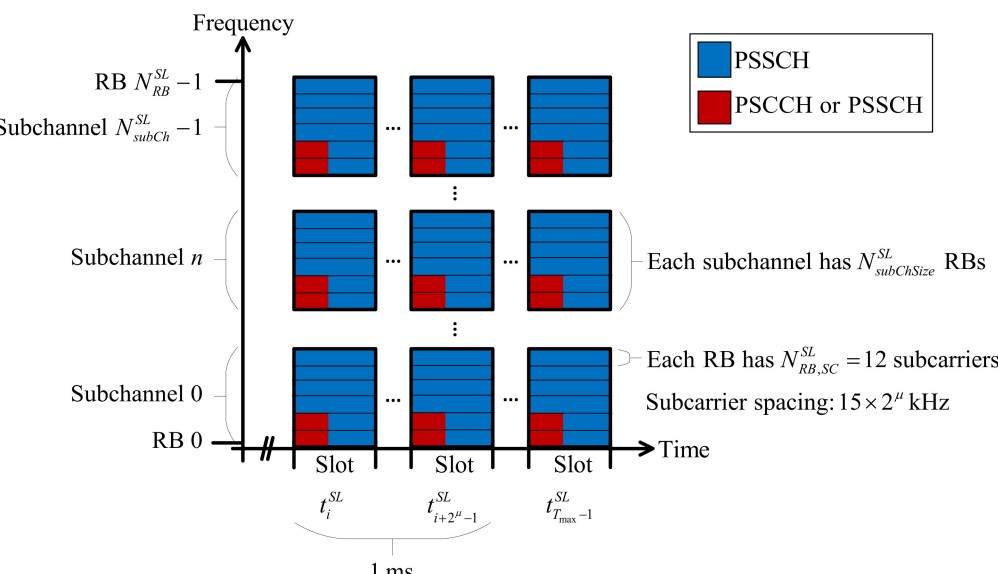

**Figure 2.** Resource grid.

The PSCCH is allocated with the corresponding PSSCH in the same slot. To enable low latency and energy efficiency, the PSCCH and PSSCH are time-division-multiplexed [28]. The allocation of PSCCH starts in the second OFDM symbol of the time slot and it can have a duration of $T_{PSCCH} \in \{2,3\}$ symbols, which are pre-configured. In the frequency domain, the PSCCH starts in the lowest RB of a subchannel, and it can occupy $N_{PSCCH} \in \{10, 12, 15, 20, 25\}$ RBs, where $N_{PSCCH} \leq N^{SL}_{subChSize}$ is also a pre-configured value [7]. The PSCCH carries the first stage SCI, i.e., the SCI format 1-A, which contains the following information [29]:

— Priority;
— Frequency resource assignment for the PSSCH;
— Time resource assignment for the PSSCH;
— Resource reservation period;
— DMRS pattern;
— Second stage SCI format;
— Beta offset indicator;
— Number of DMRS port;
— Modulation and coding scheme for the PSSCH;
— Additional Modulation and Coding Scheme (MCS) table indicator;
— Physical Sidelink Feedback Channel (PSFCH) overhead indication;
— Reserved.

Therefore, the PSSCH allocation is indicated by the SCI through the frequency resource assignment and time resource assignment fields, in a form of time RIV (TRIV) and frequency RIV (FRIV), respectively, computed as in [27]. The PSSCH allocation in a resource grid is defined in terms of starting subchannel and number of contiguously occupied subchannels. The number of contiguous subchannels occupied by a PSSCH transmission is $N_{PSSCH}$ and the starting subchannel of the first resource is $n_{start,0}$, both of which are obtained from FRIV.

There are DMRS associated to physical channels to allow the channel estimation. The DMRS associated to PSCCH for subcarrier $k$ and symbol $l$ within the slot, $d^i_{PSCCH,k,l}$, used in channel estimation are given by [30],

$$d^i_{PSCCH,k,l} = \beta_{PSCCH} w_i(k') r_l(3m + k'),\tag{1}$$

where $k = mN^{SL}_{RB,SC} + 4k' + 1$, $k' = 0, 1, 2$ and $m = 0, 1, \ldots$. The amplitude scaling factor $\beta_{PSCCH}$ is applied to transmit with the power specified in [31], and the quantity $w_i(k')$ is given by Table 2, where $i \in \{0, 1, 2\}$ shall be randomly selected [30].

**Table 2.** The quantity $w_i(k')$.

|  |  | $i$ | | |
| --- | --- | --- | --- | --- |
|  |  | 0 | 1 | 2 |
| | 0 | 1 | 1 | 1 |
| $k'$ | 1 | 1 | $e^{j\frac{2}{3}\pi}$ | $e^{-j\frac{2}{3}\pi}$ |
| | 2 | 1 | $e^{-j\frac{2}{3}\pi}$ | $e^{j\frac{2}{3}\pi}$ |

Finally, $r_l(u)$, $u = 0, 1, \ldots$ is a sequence generated by

$$r_l(u) = \frac{1}{\sqrt{2}}(1 - 2c(2u)) + j\frac{1}{\sqrt{2}}(1 - 2c(2u+1)), \tag{2}$$

where $c(u)$ is a pseudo-random sequence defined in [30].

Let be $s_q \in \mathbb{C}$, with $q = 0, 1, \ldots$, the OFDM waveform samples which carry the PSCCH/PSSCH resources allocated as in Figure 2, and the synchronization signal blocks. The transmitted signal, $\mathbf{s}_{tx,q} \in \mathbb{C}^{N_{tx}}$, is given by

$$\mathbf{s}_{tx,q} = \mathbf{f}_a s_q, \tag{3}$$

where $\mathbf{f}_a \in \mathbb{C}^{N_{tx}}$ is the precoder. Since our focus is only on the PSCCH search algorithm, we consider the simple random precoder discussed in [32], and given by

$$\mathbf{f}_a = [e^{j2\pi\omega_v}]_{1 \le v \le N_{tx}}, \tag{4}$$

where $\omega_v$, $v \in \{1, \ldots, N_{tx}\}$ are i.i.d. uniform random variables with support $\omega_v \in [0, 1]$. At the receiver side, a minimum mean-squared error (MMSE) equalizer is adopted and applied to received signal $\mathbf{s}_{rx,q} \in \mathbb{C}^{N_{rx}}$, $q = 0, 1, \ldots$, where the channel between the transmitter and receiver is estimated based on DMRS.

## 3. Proposed Effective PSCCH Searching

This section presents the proposed algorithm for the received signal processing. Firstly, the general receiver processing is introduced, and then, the commonly used Blind algorithm is discussed. After that, a brief study on the correlation of the received signal with the DMRS is presented. Based on this study, a new algorithm is proposed and finally a theoretical runtime analysis is performed.

### 3.1. Receiver Model

The block diagram of the received signal processing is shown in Figure 3. It has three parts: (1) synchronization procedures; (2) obtain control information; and (3) obtain the desired data.

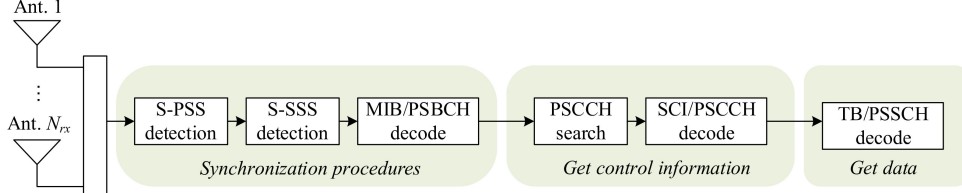

**Figure 3.** Block diagram of the receiver.

The first step of the receiver is to try to synchronize with the transmitter. For that, the receiver knows the possible Sidelink Primary Synchronization Signals (S-PSS), and performs a correlation with the received signal until it finds a significant peak. After finding the S-PSS, the receiver also performs a correlation between known sequences of the S-SSS and the received signal. Taking into account the S-PSS/S-SSS sequences that gave the

main correlation peaks, a sidelink identity is obtained, which is then used to try to decode the Master Information Block (MIB). If the MIB is successfully decoded then the receiver proceeds to the next step. Otherwise, the receiver goes back to the beginning and continues trying to synchronize.

Once the MIB is obtained, the receiver is ready to search for PSCCHs. As mentioned above, the resource grid is organized into subchannels, and the PSCCH starts in the lowest RB of a subchannel. Moreover, the DMRS associated to PSCCH are computed based on the quantity $w_i(k')$, where $i \in \{0, 1, 2\}$ shall be randomly selected. Therefore, the receiver needs to search for PSCCHs in different subchannels, and needs to try different indexes $i$ until they are able to decode the SCI/PSCCH. Thus, only after this, it is possible to get the SCI carried in the PSCCH and extract the desired data, i.e., the Transport Block (TB) from the PSSCH.

To proceed with the PSCCH search, an algorithm is needed. To the best of our knowledge the most commonly used algorithm is a blind one, as discussed in [15], whose pseudo-code is presented in the Blind algorithm (Algorithm 1).

---

**Algorithm 1:** Blind Algorithm

---

**1**　　For $n = 0, \ldots, N_{subCh}^{SL} - 1$ do
**2**　　　　For $i = 1, 2, 3$ do
**3**　　　　　　$\mathbf{c}_{n,i} = \sum\limits_{v=1}^{N_{rx}} \mathrm{corr}(\mathbf{s}_{rx,v}, \mathbf{d}_{PSCCH}^{n,i})$
**4**　　　　　　Compute $f_{offset}(q_{\max})$, where $q_{\max} = \arg\max\limits_{q \in \{0,\ldots,Q_{\max}-1\}} \mathbf{c}_{n,i}(q)$
**5**　　　　　　$\mathbf{s}_{rx,v}^{sync}(q) = \mathbf{s}_{rx,v}(q + f_{offset})$, $q = 0, \ldots, Q_{\max} - 1 - f_{offset}$
**6**　　　　　　Perform the CP-OFDM demodulation
**7**　　　　　　Extract the PSCCH candidate
**8**　　　　　　Perform the PSCCH channel estimation
**9**　　　　　　Perform the MMSE equalization
**10**　　　　　Try to decode the PSCCH
**11**　　　　　Try to decode the SCI
**12**　　　　　If SCI decoded with success, then
**13**　　　　　　　　Break
**14**　　　　End
**15**　　End
**16**　End

---

As we can see in the Blind algorithm, we search in all subchannels (line 1), for all indexes $i$ (line 2) until decode the SCI/PSCCH (lines 12–14). In the PSCCH decoding processing, first we compute the correlation of received signal by each antenna, $\mathbf{s}_{rx,v} = [s_{rx,v}(0), \ldots, s_{rx,v}(Q_{\max} - 1)]^T$, $v = 1, \ldots N_{rx}$, where $Q_{\max}$ is the maximum number of samples used in the correlation, with the DMRS, $\mathbf{d}_{PSCCH}^{n,i} = \left[ \ldots, d_{PSCCH,k,l}^{i}, \ldots \right]^T$, $k = nN_{subChSize}^{SL} N_{RB,SC}^{SL} + 4k' + 1$, $k' = 0, 1, 2$, $l = 1, \ldots, T_{PSCCH}$ (line 3). Then, the frame offset, $f_{offset}(q_{\max})$, is computed using the index number of sample where the correlation peak is stronger (line 4) and a subframe synchronization is completed (line 5). After that, the CP-OFDM demodulation is performed (line 6), the PSCCH candidate for subchannel $n$ is extracted (line 7), the channel estimation using $\mathbf{d}_{PSCCH}^{n,i}$ is performed (line 8), and then the MMSE equalization is done using the estimated channel (line 9). Finally, we try to decode the PSCCH and SCI (lines 10–11).

### 3.2. Correlation Analysis

An analysis of the correlations performed in the Blind algorithm, line 3, is completed in this Section. If the received signal has a PSCCH present in a subchannel $n$ with DMRS using an index $i$, it would be expected to have a stronger correlation peak when $\mathrm{corr}(\mathbf{s}_{rx,v}, \mathbf{d}_{PSCCH}^{n,i})$ is conducted. We observed it by transmitting a signal with a PSCCH on the subchannel $n = 1$ and DMRS using the index $i = 1$. This test is conducted for a

scenario where the signal-to-noise-ratio (SNR) is set to $-10$ dB. The remaining parameters are $N_{RB}^{SL} = 50$, $N_{subChSize}^{SL} = 10$, $N_{subCh}^{SL} = 5$, $N_{PSCCH} = 10$, $T_{PSCCH} = 3$, $N_{PSSCH} = 1$, $N_{tx} = 16$, and $N_{rx} = 16$. The results are shown in Figure 4.

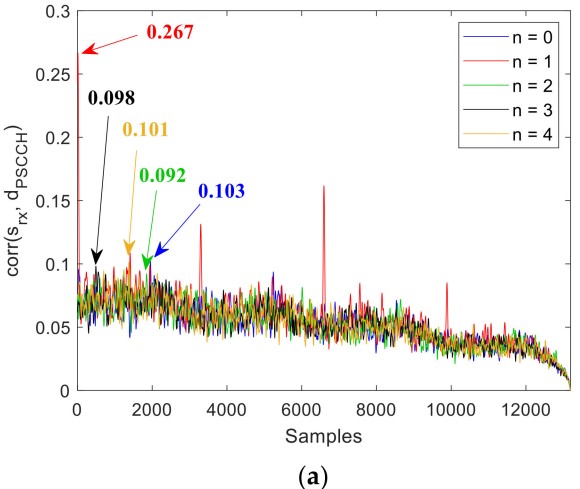

(a)

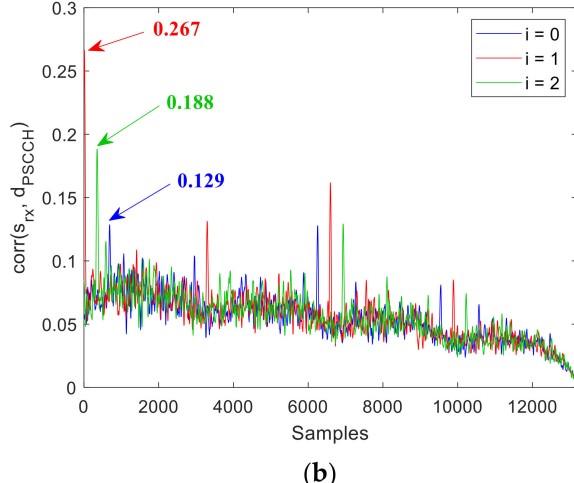

(b)

**Figure 4.** Correlation results: (**a**) Trying different subchannels with $i = 1$; (**b**) Trying in subchannel $n = 1$ for indexes $i = 0, 1$ and 2.

As we can see in Figure 4a, the peak associated to the right subchannel, 0.267 is much higher than the other peaks associated to other subchannels, 0.103, 0.101, 0.098, and 0.092. Thus, the use of this information can be very effective in determining the right subchannel. Moreover, we can also see in Figure 4b, that the peak associated to the right index $i$ is higher than the other peaks, and thus we can also use this information in determining the right index $i$.

### 3.3. Proposed Effective PSCCH Searching Based on Correlation

Based on the observations of the previous section, we now design a new efficient algorithm, where all correlations are firstly completed, without performing all the remaining process. After that, the processing for SCI/PSCCH search and decoding are completed for the pair $(n, i)$, where the correlation is stronger and it continues in descending order. The pseudo-code of the Proposed algorithm (Algorithm 2) is presented following.

---

**Algorithm 2:** Proposed Effective PSCCH Searching Algorithm Based on Correlation

| | |
|---|---|
| 1 | For $n = 0, \ldots, N_{subCh}^{SL} - 1$ do |
| 2 | For $i = 1, 2, 3$ do |
| 3 | $\mathbf{c}_{n,i} = \sum\limits_{v=1}^{N_{rx}} \text{corr}(\mathbf{s}_{rx,v}, \mathbf{d}_{PSCCH}^{n,i})$ |
| 4 | $\mathbf{c}^{\max}(3n + i - 1) = \max(\mathbf{c}_{n,i})$ |
| 5 | End |
| 6 | End |
| 7 | $(\mathbf{c}^{sorted}, \mathbf{c}_{indexes}^{sorted}) = \text{sort}_{desc}(\mathbf{c}^{\max})$ |
| 8 | For $p = 0, \ldots, 3N_{subCh}^{SL} - 1$ do |
| 9 | $n = \mathbf{c}_{indexes}^{sorted}(p)/3$ |
| 10 | $i = \text{rem}\left(\mathbf{c}_{indexes}^{sorted}(p), 3\right) - 1$ |
| 11 | Compute $f_{offset}(q_{\max})$, where $q_{\max} = \arg\max\limits_{q \in \{0, \ldots, Q_{\max} - 1\}} \mathbf{c}_{n,i}(q)$ |
| 12 | $\mathbf{s}_{rx,v}^{sync}(q) = \mathbf{s}_{rx,v}(q + f_{offset})$, $q = 0, \ldots, Q_{\max} - 1 - f_{offset}$ |
| 13 | Perform the CP-OFDM demodulation |
| 14 | Extract the PSCCH candidate |

---

| Algorithm 2: *Cont.* | | |
|---|---|---|
| **15** | | Perform the PSCCH channel estimation |
| **16** | | Perform the MMSE equalization |
| **17** | | Try to decode the PSCCH |
| **18** | | Try to decode the SCI |
| **19** | | If SCI decoded with success, then |
| **20** | | Break |
| **21** | | End |
| **22** | End | |

In the Proposed algorithm, first we compute for all subchannels (line 1) and all indexes $i$ (line 2), the correlations $\text{corr}(\mathbf{s}_{rx,v}, \mathbf{d}_{PSCCH}^{n,i})$ (line 3), and store the value of main peak associated to each pair $(n, i)$ in $\mathbf{c}^{\max} \in \mathbb{R}^{3N_{subCh}^{SL}}$ (line 4). After that, we sort the vector $\mathbf{c}^{\max}$ in descending order and store the indexes of original positions in $\mathbf{c}_{indexes}^{sorted} \in \mathbb{Z}^{3N_{subCh}^{SL}}$ (line 7). Then, for each element of $\mathbf{c}_{indexes}^{sorted}$, we get the pair $(n, i)$ (lines 9–10), and we apply the remain processing (lines 11–21), similar to the Blind algorithm.

### 3.4. Theoretical Runtime Analysis

In this section, a theoretical runtime analysis is completed, in order to compare the performances of the two presented algorithms, i.e., the conventional Blind algorithm, and the Proposed algorithm. For that, first let us define the processing time of each part of the algorithms. These definitions are presented in Table 3.

**Table 3.** Runtime definitions.

| Symbol | Designation | Rows of Blind Algorithm | Rows of Proposed Algorithm |
|---|---|---|---|
| $T_{corr}$ | Compute correlation | 3 | 3 |
| $T_{fmax}$ | Find the maximum correlation | - | 4 |
| $T_{sort}$ | Sort the maximum correlations | - | 7 |
| $T_{gpair}$ | Get the pair $(n, i)$ | - | 9–10 |
| $T_{rem}$ | Remain processing | 4–14 | 11–21 |

Let $E_1$ and $E_2$ be the expected number of attempts associated to Blind and Proposed algorithms, respectively. Therefore, the total runtime needed to perform the search and decoding procedure of an SCI/PSCCH, associated to Blind and Proposed algorithms, are given, respectively, by

$$T_1 = E_1(T_{corr} + T_{rem}), \tag{5}$$

$$T_2 = 3N_{subCh}^{SL}\left(T_{corr} + T_{fmax}\right) + T_{sort} + E_2\left(T_{gpair} + T_{rem}\right). \tag{6}$$

For a low SNR scenario, $E_1 = E_2 = 3N_{subCh}^{SL}$, and from Equations (5) and (6) we obtain

$$T_1 = 3N_{subCh}^{SL}(T_{corr} + T_{rem}), \tag{7}$$

$$T_2 = 3N_{subCh}^{SL}\left(T_{corr} + T_{fmax} + T_{gpair} + T_{rem}\right) + T_{sort}. \tag{8}$$

Since $T_{fmax}$, $T_{gpair}$ and $T_{sort}$ are much lower than $T_{corr}$ or $T_{rem}$, for a low SNR scenario, we conclude that $T_1 \approx T_2$.

For a high SNR scenario, $E_1 = \frac{1}{2}\left(3N_{subCh}^{SL} + 1\right)$ and $E_2 = 1$. Then, from Equations (5) and (6) we obtain

$$T_1 = 1.5N_{subCh}^{SL}T_{corr} + 1.5N_{subCh}^{SL}T_{rem} + 0.5T_{corr} + 0.5T_{rem}, \tag{9}$$

$$T_2 = 3N_{subCh}^{SL}T_{corr} + T_{rem} + 3N_{subCh}^{SL}T_{fmax} + T_{sort} + T_{gpair}. \tag{10}$$

Since $T_{fmax}$, $T_{gpair}$ and $T_{sort}$ are much lower than $T_{corr}$ or $T_{rem}$, then $T_2 \approx 3N_{subCh}^{SL}T_{corr} + T_{rem}$. Therefore, the Proposed algorithm has a clear advantage over the Blind algorithm when $T_2 < T_1 \Leftrightarrow T_{corr} < T_{rem}$, which is generally true, since $T_{rem}$ includes CP-OFDM demodulation, PSCCH channel estimation, MMSE equalization, and the SCI/PSCCH decoding.

For instance, in a machine with Intel(R) Core(TM) i7-10750H CPU @ 2.60GHz and 16.0 GB of RAM, we obtain for the scenario of Results section with $N_{subCh}^{SL} = 5$, the values $T_{corr} = 20,285\,\mu s$, $T_{fmax} = 34\,\mu s$, $T_{sort} = 9\,\mu s$, $T_{gpair} = 6\,\mu s$, and $T_{rem} = 96,543\,\mu s$. Then, from (9) and (10) we have $T_1 = 934,624\,\mu s$ and $T_2 = 401,343\,\mu s$.

## 4. Results

In this section, we compare the performance of the Blind and Proposed algorithms. For that, we assume a scenario where $N_{subChSize}^{SL} = 10$, $N_{PSCCH} = 10$, $T_{PSCCH} = 3$, $N_{PSSCH} = 1$, $N_{tx} = 16$ and $N_{rx} = 16$. The PSCCH is transmitted in a subchannel $n \in \{0, \ldots, N_{subCh}^{SL} - 1\}$ selected randomly, and associated to DMRS with a random index $i \in \{0, 1, 2\}$. We tested with $N_{RB}^{SL} = 50$ and $N_{RB}^{SL} = 100$, therefore $N_{subCh}^{SL} = N_{RB}^{SL}/N_{subChSize}^{SL}$ is 5 or 10. Finally, the results were obtained on a machine with Intel(R) Core(TM) i7-10750H CPU @ 2.60 GHz and 16.0 GB of RAM.

Figure 5 shows the comparision of the SCI BLER performance for Blind and Proposed algorithms. For both cases, almost all blocks failed to decode until SNR $= -18\,dB$ and almost all blocks were successful when decoding above SNR $= -12\,dB$. So, there was a significant improvement going from $-18$ to $-12\,dB$. Moreover, we can observe that the performance obtained for the two algorithms almost overlaps.

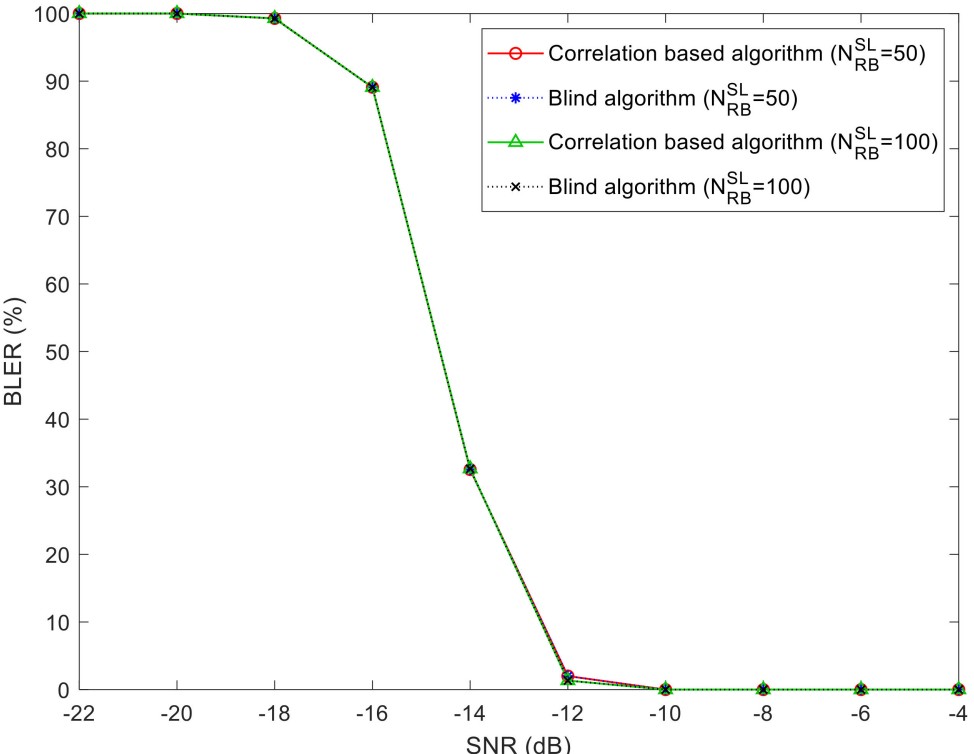

**Figure 5.** SCI BLER.

Figure 6 shows the throughput obtained for both algorithms. As we can see, the throughput is very low until SNR $= -12\,dB$ and it is almost 100% when above SNR $= -8\,dB$. Therefore, we also have a sudden performance improvement, although this only occurs for slightly higher SNR, when comparing with the results in Figure 5. The throughput obtained for the two algorithms practically overlaps.

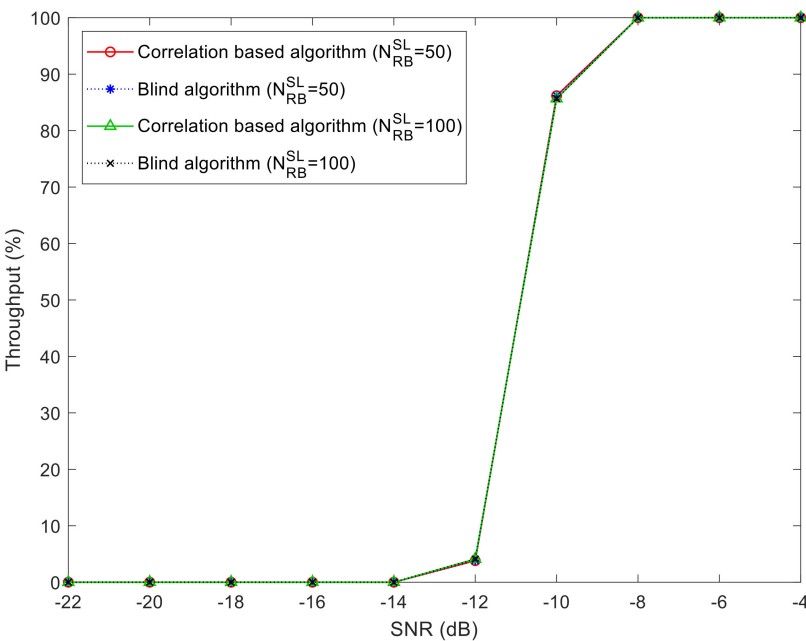

**Figure 6.** Throughput for PSSCH.

Figure 7 shows the average number of attempts for the Blind and Proposed algorithms, for $N_{RB}^{SL} = 50$ and $N_{RB}^{SL} = 100$ ($N_{subCh}^{SL} = 5$ and $N_{subCh}^{SL} = 10$, respectively). As mentioned above, for low SNR, $E_1 = E_2 = 3N_{subCh}^{SL}$. Then, for $N_{RB}^{SL} = 50$ we have $E_1 = E_2 = 15$, and for $N_{RB}^{SL} = 100$ we have $E_1 = E_2 = 30$, which is validated in Figure 7. Moreover, for high SNR, $E_1 = \frac{1}{2}\left(3N_{subCh}^{SL} + 1\right)$ and $E_2 = 1$. Then, for $N_{RB}^{SL} = 50$ we have $E_1 = 8$, and for $N_{RB}^{SL} = 100$ we have $E_1 = 15.5$, which is also validated in Figure 7. As shown in Figures 5 and 6, the same performance in terms of BLER and throughput was achieved, but with a much lower number of attempts to decode the PSCCH at it can be seen in Figure 7. Similar to Figure 5, almost all attempts failed until SNR $= -18$ dB and the best performance is achieved above SNR $= -12$ dB.

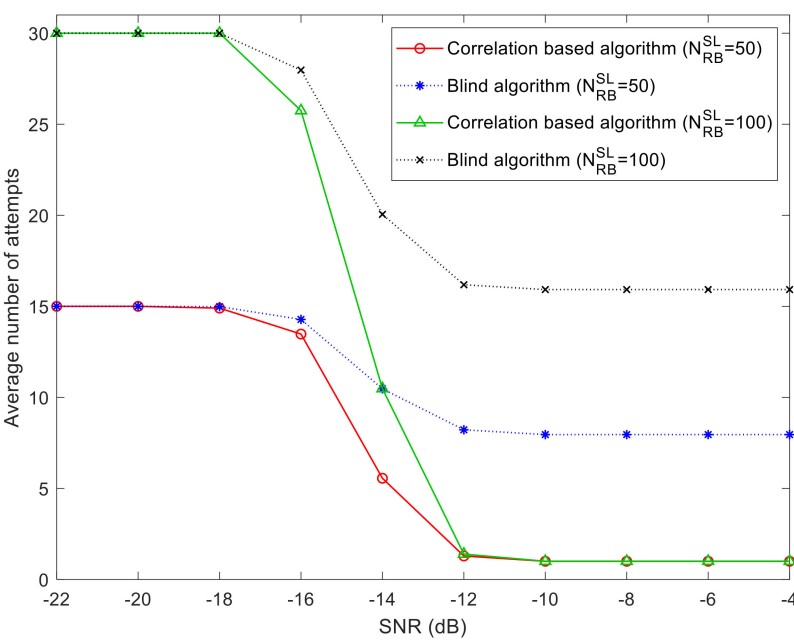

**Figure 7.** Average number of attempts.

Figure 8 shows the runtime measures for PSCCH search/decoding. As expected, the correlation-based algorithm is better for high SNR regime, when compared with the correspondent Blind algorithm result. In the high SNR regime, we have for $N_{RB}^{SL} = 50$, a total runtime $T_1 = 0.9\,\text{s}$, and $T_2 = 0.4\,\text{s}$, which are approximately the same values as the ones obtained in Section 3.4, and for $N_{RB}^{SL} = 100$, we have $T_1 = 3.0\,\text{s}$ and $T_2 = 1.6\,\text{s}$. Therefore, in both cases the runtime savings were considerable.

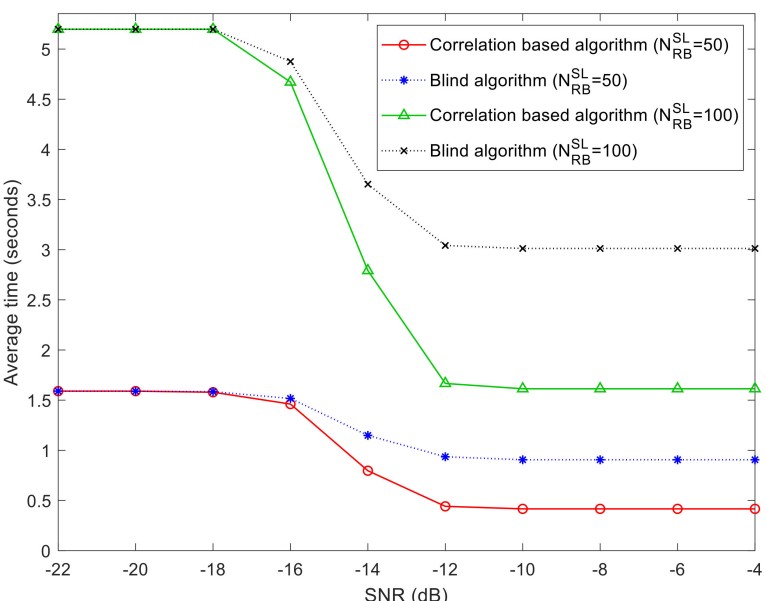

**Figure 8.** Average time for PSCCH search/decoding (seconds).

Finally, Figure 9 shows the same results of Figure 8, but in a normalized way to the highest time, so that we can analyze the results without these being associated with a specific CPU. In the high SNR zone, the gap for $N_{RB}^{SL} = 50$ is 30.8%, while for $N_{RB}^{SL} = 100$ it is 26.9%. This means that the proposed correlation-based algorithm, even though it continues to show a clear improvement over the Blind algorithm, lost some effectiveness by increasing the $N_{RB}^{SL}$. This is associated with the fact that the bandwidth is greater, and that more samples are being processed in calculating the correlations.

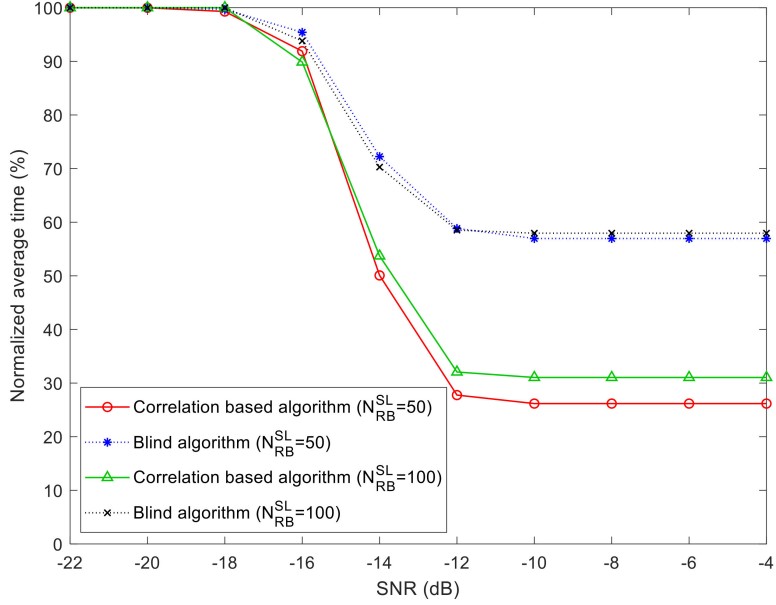

**Figure 9.** Normalized average time for PSCCH search/decoding (%).

## 5. Conclusions

In this paper, we developed an effective PSCCH searching algorithm for 5G-NR V2X sidelink communications. In the proposed approach, we firstly computed all the correlations between the received signal and the PSCCH DMRS, and the remaining conventional processing to decode the PSCCH was only performed over the subchannels with higher correlation. The Proposed algorithm was compared with the conventional Blind algorithm that ensured the best possible performance in terms of PSCCH detection.

The results showed that the proposed algorithm achieves the same performance in terms of SCI BLER and throughput, but with a much shorter runtime. For $N_{RB}^{SL} = 50$, the runtime of the Proposed algorithm was less than half of the Blind algorithm, which is an important improvement because the PSCCH detection and decoding have a very significant weight in the overall processing of the receiver. Furthermore, the average number of attempts to decode the PSCCH always tends to be one for high SNR, regardless of bandwidth. Thus, the proposed PSCCH searching algorithm is efficient for very high bandwidths, which are expected in 5G-NR communications. However, it would be interesting in future work to reduce the runtime in the calculation of correlations, which has significant weight in the Proposed algorithm.

Finally, the Proposed algorithm could clearly replace the blind approaches for PSCCH searching in all schemes presented in [15–26]. Therefore, it could be an interesting approach for real-time systems, since it ensures a significant reduction of runtime and it is compatible with several schemes proposed in the literature.

**Author Contributions:** Investigation, J.D. and R.M.; Supervision, A.S., P.M. and R.M.; Validation, A.S., P.M. and R.M.; Writing—original draft, J.D. and R.M.; Writing—review and editing, A.S. and P.M. All authors have read and agreed to the published version of the manuscript.

**Funding:** This work was financially supported by Fundo Europeu de Desenvolvimento Regional (FEDER) through the POCI-01-0247-FEDER-046962 (5GAUTO) and by FCT/MCTES through national funds and when applicable co-funded EU funds under the project UIDB/50008/2020-UIDP/50008/2020.

**Conflicts of Interest:** The authors declare no conflict of interest.

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
