# Peer review of "Effective PSCCH Searching for 5G-NR V2X Sidelink Communications"

_electronics, doi:10.3390/electronics10222827_

Round 1

Reviewer 1 Report

The paper contains the comparison between a blind algorithm and their proposed algorithm for PSCCH in 5G-NR V2X sidelink communications. For that, authors provided a simulation to bring out the their algorithm performance compared to other. 

The paper is a solid work with good technical contents with a completeness and accuracy of references and figures. The importance and timeliness of the topic addressed in the paper within its area of research is good. 

The introduction section is clear, but it is not explained the motivation to select blind algorithm for compared study. The blind algorithm is used in references 14, but in section 1.1 there are 11 references. I suggest to add a motivation of their choice and a table to compare the references 14-25, used in section 1.1.

The mathematical treatment is clear, but some superscripts and subscripts are not properly abbreviated or the abbreviations are not explained, e.g. in row 180 N^{SL}_{RB,SC}, SL and SC are not explained.

In receiver design section, authours used the terms "algorithm 1" to define blind alogrithm and "algorithm 2" to define their proposed algorithm. The use of numbers to define two algorithms is not appropriate in scientific work. I suggest to used, for example, "blind algorithm" and "proposed algorithm". 

The results section is clear, but the conclusion section is too short. There are no future developments and possible benefit in application fields.

Author Response

Dear Reviewer,

This is the revised version of the journal "Effective PSCCH searching for 5G-NR V2X sidelink communications ( Manuscript ID electronics-1460963)" submitted for possible publication in Electronics.

First of all, we would like to say many thanks for the valuable comments and suggestions, which clearly allowed us to improve this paper. We believe all issues raised were properly addressed in the revised version. In the following, we have detailed answers to those issues.

Best regards

The authors

Reply to Reviewer-1

Reviewer Comment-1: The paper contains the comparison between a blind algorithm and their proposed algorithm for PSCCH in 5G-NR V2X sidelink communications. For that, authors provided a simulation to bring out the their algorithm performance compared to other. The paper is a solid work with good technical contents with a completeness and accuracy of references and figures. The importance and timeliness of the topic addressed in the paper within its area of research is good. The introduction section is clear, but it is not explained the motivation to select blind algorithm for compared study. The blind algorithm is used in references 14, but in section 1.1 there are 11 references. I suggest to add a motivation of their choice and a table to compare the references 14-25, used in section 1.1.

Authors’ reply: Many thanks for your positive and constructive comments. We agree with the reviewer that would be interesting to add a motivation for our choice, as well as a table to compare the references. We added the Table 1 in 1.1, and the following text in a new section 1.2: “Although there are several algorithms to reserve resources for transmission as presented above, in the receiving UE, the PSCCH searching is done in a pure blind way, trying all the possibilities, or at most is considered some kind of threshold to try to eliminate candidates. The threshold approach is not efficient because if the threshold is low, we still have a high number of candidates, or if the threshold is high, we could be eliminating candidates that contained a PSCCH and thus deteriorating the overall system performance. In a semi-persistent subchannel selection we can perform an optimization in the PSCCH searching because allocation is constant over a period of time, however this approach can lead to persistent collisions during this period, mainly in the Mode 2 of the 5G-NR V2X [14]. Therefore, the blind algorithm is the one that guarantees us the best system performance in terms of data reliability, but at the expense of longer runtime as all possibilities are tested. Then, a new algorithm with the same performance as the Blind algorithm, but with more efficient processing to reduce the runtime is crucial for practical 5G-NR V2X systems.”

********

Reviewer Comment-2: The mathematical treatment is clear, but some superscripts and subscripts are not properly abbreviated or the abbreviations are not explained, e.g. in row 180 N^{SL}_{RB,SC}, SL and SC are not explained.

Authors’ reply: Following the reviewer suggestion, we added the meaning of SL and SC. Now, the sentence is: “The smallest unit in a sidelink (SL) communication is the RE, composed by a subcarrier k, and an OFDM symbol l, where  subcarriers (SC) are a RB, with a subcarrier spacing equal to (…)”

********

Reviewer Comment-3: In receiver design section, authours used the terms "algorithm 1" to define blind alogrithm and "algorithm 2" to define their proposed algorithm. The use of numbers to define two algorithms is not appropriate in scientific work. I suggest to used, for example, "blind algorithm" and "proposed algorithm".

Authors’ reply: Thank you for pointing out this issue. Following the reviewer’s suggestion, we modified the algorithm’s name.

********

Reviewer Comment-4: The results section is clear, but the conclusion section is too short. There are no future developments and possible benefit in application fields.

Authors’ reply: Thank you for your suggestion. We improved the conclusion section and we believe which is more complete now.

Reviewer 2 Report

The paper  proposes an effective PSCCH searching algorithm for 5G-NR V2X sidelink communications.

I would like to suggest the following comments to be addressed:

  • The motivation of the work needs to be elaborated
  • in second bullet of contributions, "based on the previous results ... " is not clear. What results the authors are talking about? I suggest the entire contribution section to be rewritten.
  • In system model, the overview of the architecture is not clear.
  • section 3 is called "receiver design" but shouldn't be "the proposed algorithm"? or something  that shows your contribution?
  • explain the algorithm with an example for more clarity
  • What would be the limitations of your work to be addressed in the future work?

I would also suggest the following references:

G. Naik, B. Choudhury and J. Park, "IEEE 802.11bd & 5G NR V2X: Evolution of Radio Access Technologies for V2X Communications," in IEEE Access, vol. 7, pp. 70169-70184, 2019, doi: 10.1109/ACCESS.2019.2919489.

K. Ganesan, P. B. Mallick, J. Löhr, D. Karampatsis and A. Kunz, "5G V2X Architecture and Radio Aspects," 2019 IEEE Conference on Standards for Communications and Networking (CSCN), 2019, pp. 1-6, doi: 10.1109/CSCN.2019.8931319.

A. Ghosh, A. Maeder, M. Baker and D. Chandramouli, "5G Evolution: A View on 5G Cellular Technology Beyond 3GPP Release 15," in IEEE Access, vol. 7, pp. 127639-127651, 2019, doi: 10.1109/ACCESS.2019.2939938.

Author Response

Dear Reviewer,

This is the revised version of the journal "Effective PSCCH searching for 5G-NR V2X sidelink communications ( Manuscript ID electronics-1460963)" submitted for possible publication in Electronics.

First of all, we would like to say many thanks for the valuable comments and suggestions, which clearly allowed us to improve this paper. We believe all issues raised were properly addressed in the revised version. In the following, we have detailed answers to those issues.

Best regards

The authors

Reply to Reviewer-2

Reviewer Comment-1: The motivation of the work needs to be elaborated

Authors’ reply: Following the reviewer suggestion, we added the following text in a new section 1.2 (new section in this new version): “Although there are several algorithms to reserve resources for transmission as presented above, in the receiving UE, the PSCCH searching is done in a pure blind way, trying all the possibilities, or at most is considered some kind of threshold to try to eliminate candidates. The threshold approach is not efficient because if the threshold is low, we still have a high number of candidates, or if the threshold is high, we could be eliminating candidates that contained a PSCCH and thus deteriorating the system performance. In a semi-persistent subchannel selection we can perform an optimization in the PSCCH searching because allocation is constant over a period of time, however this approach can lead to persistent collisions during this period, mainly in the Mode 2 of the 5G-NR V2X [14]. Therefore, the blind algorithm is the one that guarantees us the best system performance in terms of data reliability, but at the expense of longer runtime as all possibilities are tested. Then, a new algorithm with the same performance as the Blind algorithm, but with more efficient processing to reduce the runtime is crucial for practical 5G-NR V2X systems.”

********

Reviewer Comment-2: in second bullet of contributions, "based on the previous results ... " is not clear. What results the authors are talking about? I suggest the entire contribution section to be rewritten.

Authors’ reply: Thank you for pointing out these issues. We improved the contribution section and we believe that now is clearer.

********

Reviewer Comment-3: In system model, the overview of the architecture is not clear.

Authors’ reply: Thank you for your commentary. In order to improve the system model, some sentences have been clarified, and more details have been added in Figure 1.

********

Reviewer Comment-4: section 3 is called "receiver design" but shouldn't be "the proposed algorithm"? or something  that shows your contribution?

Authors’ reply: Many thanks for your suggestion. Now, section 3 is called: “Proposed effective PSCCH searching”.

********

Reviewer Comment-5: explain the algorithm with an example for more clarity

Authors’ reply: Thank you for your suggestion. However, we would like to point out that we presented the pseudo-code and then explained it in detail line by line. To explain the algorithm with an example, for instance with the parameters used in Results section, we would only repeat everything that was said before, only replacing  by 5 or 10. It is an algorithm to reduce runtime, then the example we can present is the runtimes of each part of the algorithms. Therefore, we added in section 3.4: “For instance, in a machine with Intel(R) Core(TM) i7-10750H CPU @ 2.60GHz and 16.0 GB of RAM, we get for the scenario of Results section with , the values , , , , and . Then, from (9) and (10) we have  and .” Hope you understand our point of view.

********

Reviewer Comment-6: What would be the limitations of your work to be addressed in the future work?

Authors’ reply: Many thanks for that very pertinent question. We added in the Conclusions section: “However, it would be interesting in future work to reduce the runtime in the calculation of correlations, which has significant weight in the Proposed algorithm.”

********

Reviewer Comment-7: I would also suggest the following references:

  1. Naik, B. Choudhury and J. Park, "IEEE 802.11bd & 5G NR V2X: Evolution of Radio Access Technologies for V2X Communications," in IEEE Access, vol. 7, pp. 70169-70184, 2019, doi: 10.1109/ACCESS.2019.2919489.

  1. Ganesan, P. B. Mallick, J. Löhr, D. Karampatsis and A. Kunz, "5G V2X Architecture and Radio Aspects," 2019 IEEE Conference on Standards for Communications and Networking (CSCN), 2019, pp. 1-6, doi: 10.1109/CSCN.2019.8931319.

  1. Ghosh, A. Maeder, M. Baker and D. Chandramouli, "5G Evolution: A View on 5G Cellular Technology Beyond 3GPP Release 15," in IEEE Access, vol. 7, pp. 127639-127651, 2019, doi: 10.1109/ACCESS.2019.2939938.

Authors’ reply: Many thanks for the suggestions. They are interesting references, one of them was already referenced in the paper, but now we have also added the other two references.

Round 2

Reviewer 1 Report

New version of manuscript is more clear than previous version. Authors followed the reviewers suggestions.